# Relevant *Fusarium* Mycotoxins in Malt and Beer

**DOI:** 10.3390/foods11020246

**Published:** 2022-01-17

**Authors:** Xenia Pascari, Sonia Marin, Antonio J. Ramos, Vicente Sanchis

**Affiliations:** AGROTECNIO-CERCA Center, Applied Mycology Unit, Food Technology Department, University of Lleida, Av. Rovira Roure 191, 25198 Lleida, Spain; xenia.pascari@udl.cat (X.P.); sonia.marin@udl.cat (S.M.); antonio.ramos@udl.cat (A.J.R.)

**Keywords:** beer, craft beer, *Fusarium*, mycotoxins, brewing

## Abstract

Mycotoxins are secondary fungal metabolites of high concern in the food and feed industry. Their presence in many cereal-based products has been numerously reported. Beer is the most consumed alcoholic beverage worldwide, and *Fusarium* mycotoxins originating from the malted and unmalted cereals might reach the final product. This review aims to describe the possible *Fusarium* fungi that could infect the cereals used in beer production, the transfer of mycotoxins throughout malting and brewing as well as an insight into the incidence of mycotoxins in the craft beer segment of the industry. Studies show that germination is the malting step that can lead to a significant increase in the level of all *Fusarium* mycotoxins. The first step of mashing (45 °C) has been proved to possess the most significant impact in the transfer of hydrophilic toxins from the grist into the wort. However, during fermentation, a slight reduction of deoxynivalenol, and especially of zearalenone, is achieved. This review also highlights the limited research available on craft beer and the occurrence of mycotoxins in these products.

## 1. Introduction

*Fusarium* is a fungal genus belonging to the phylum Ascomycota and comprising more than 1500 species [1]. Most of them are essential to the environment, but some present certain pathogenicity to animals and humans by producing mycotoxins. Species such as *Fusarium graminearum* Schwabe, *F. oxysporum* and *F. verticillioides* are very common plant pathogens.

Beer is a carbonated fermented beverage obtained from malted cereals. The brewing tradition is one of the oldest ones known to civilized humanity, with historical evidence suggesting it began no earlier than 5000 BC [2]. Barley (*Hordeum vulgare*) is by far the most common cereal used to produce beer; however, in some countries, beer can also be obtained from malted wheat, rye, or sorghum. Although the brewing process is not as efficient as with barley, celiac-friendly beers from millet or buckwheat are becoming increasingly available on the market as well. Cereals such as rice or maize are primarily used as adjuncts by some brewers, depending on which one best fits the chosen process. 

Both brewing grains and adjuncts are carefully selected for quality prior to their use. Parameters such as water content (<14.5%), germination energy (>95%), protein content (9.5 to 11.5%), minimum weathering and microbial count, are just a few to consider [3]. Contamination with *Fusarium* or other toxigenic or non-toxigenic fungi can greatly interfere with the plants’ metabolism and therefore alter the composition of the grains and brewing-related enzymes, besides influencing the safety of the final product [4]. Among the most occurring *Fusarium* mycotoxins recorded in beer are zearalenone (ZEN), type B trichothecenes as nivalenol (NIV), deoxynivalenol (DON), deoxynivalenol-3-glucoside (DON-3-Glc), 3- and 15-acetyl-deoxynivalenol (3Ac- and 15-Ac-DON) and, to a lesser extent, type A trichothecenes such as T-2 and HT-2 toxins [5]. Their intake is associated with many acute and chronic toxic effects in both animals and humans (Table 1). The available studies focus on the toxicity of mycotoxins and analysis techniques [6] and identifying the incidence of these mycotoxins in beer or raw materials [7,8,9,10,11], while others study how the mycotoxins found in the cereals can be transferred to the final product and which steps are of the utmost importance [12,13,14,15]. Surveys are an important source for estimating the risk of exposure to mycotoxin associated with certain foods; nonetheless, in the case of beer, there are only annual data on the consumption patterns of these products, which makes the estimation of the contribution of these products to the daily exposure challenging [16]. A recent survey estimating the exposure to mycotoxins through the consumption of alcoholic beverages identified that the 40 beer samples analysed contained at least one mycotoxin [17]. Among them, alternariol (AOH) and DON were the most prevalent in beer with an average concentration of 24.9 and 8.65 µg/L, respectively. Azam et al., (2021) [18] reviewed not only the mycotoxins occurring in different beers and other beverages but also today’s strategies for their detection and mitigation. A very detailed systematic review and meta-analysis of mycotoxins in beer were recently published [19]. The authors estimated the contribution of beer consumption to the tolerable daily intake (TDI) or provisional maximum tolerable daily intake (PMTDI) of different *Fusarium* mycotoxins in different countries and regions worldwide. In the case of Spain, they identified that beer consumption by a person with a bodyweight of 70.8 kg will contribute with 3.58% of the PMTDI for DON and its derivatives (1 µg/kg body weight/day), 7.34% of the TDI for fumonisins (1 µg/kg body weight/day) and 9.73% of the PMTDI for ZEN and its derivatives (0.5 µg/kg body weight/day).

The present work aims to review the available literature regarding the impact of the presence of *Fusarium* genera in the barley-to-beer chain. It will investigate the extent of mycotoxin contamination in brewing cereals, the transfer of *Fusarium* mycotoxins from raw materials to the final product in the case of beers obtained from pale malts, revise the problem of mycotoxins in the case of craft beer production and provide several future perspectives. 

## 2. *Fusarium* Fungi in Brewing Cereals

*Fusarium* is a filamentous fungus, introduced for the first time by Link in the year 1809 as *Fusisporium* (Figure 1) [33]. It can produce a vast number of plant diseases, including root or stem rots, cankers, wilts, fruit or seed rots and leaf diseases [1]. *Fusarium* infestation is not limited to a particular region, being equally difficult to control in areas with a temperate climate as well as in tropical areas [34]. Cereal crops are mainly affected by grain and seed blights, most often caused by *F. graminearum* and *F. culmorum* in wheat and barley, *F. verticillioides* in maize and *F. thapsinum* in sorghum, leading to yield losses and mycotoxins production [34]. Besides the type of cereal, the adopted agricultural practices together with the weather conditions over each year greatly define the *Fusarium* population to be developed in each geographical region [35]. Some researchers also suspect a considerable modification in both the fungal profile and plants’ response to the infection due to climate change [36]. 

The following subsections will describe in more detail the problems that can result from using *Fusarium*-infected cereals in beer production. 

### 2.1. Barley

Barley (*Hordeum vulgare* L.) is an important cereal crop. According to the European Commission (2021), it was characterized by a production of 55.6 million tonnes in 2019 in the European Union only. To be suitable for the brewing industry it must fulfil several conditions, such as a high germination capacity, low protein content, high malt extract and diastatic power, low colour level and a uniformity of the grain size. Kaur, Bowman, Stewart and Evans (2015) [37] studied how the fungal community of barley malts from different geographical regions correlated with the quality parameters of these malts. They identified significant differences in the fungal population of barley from South Africa and the countries of the Northern Hemisphere, both quantitatively (abundance of the fungi) and qualitatively (type of fungi). These differences were significantly correlated with the usual quality parameters checked in malting barley. Their study also complemented the results obtained by Schwarz et al. (2002) [38] that indicated a strong association between the *Fusarium* fungi implicated in FHB (*Fusarium* head blight) in barley and increased activity of the ß-glucanase, xylanase and proteinase activities in the grain–events translated into decreased malt yield, wort ß-glucans and viscosity, and increased wort soluble nitrogen and free amino nitrogen (FAN) and beer gushing (excessive foam formation without previously shaking the bottle). 

*F. graminearum* was commonly considered the main species isolated from FHB cereals; however, *F. poae* has been also increasingly found in recent years’ surveys [39]. Both can produce trichothecenes, proved to be responsible for the aggressiveness or virulence of the fungi in the barley plant [40]. DON is the most common mycotoxin in malting barley, followed by ZEN, T-2 and HT-2 toxins [41,42]. A survey to analyse fungal metabolites, including mycotoxins, in 253 barley samples from the crop season 2016–2017 in Switzerland, performed by Drakopoulos, Sulyok, Krska, Logrieco and Vogelgsang (2021) [43], shows that emerging mycotoxins such as enniatins and beauvericin are the most common mycotoxins in barley grains (between 70% and 100% of positive samples). Although DON presented a higher occurrence rate (69%) compared to DON-3-Glc (39%), the average concentration levels of DON-3-Glc were almost 7 times higher compared to that of DON (909 µg/kg and 7030 µg/kg in DON and DON-3Glc, respectively).

New varieties of malting barley resistant to FHB are constantly registered [44,45], with up to 50% less DON accumulated in the kernels compared to a similar non-resistant variety. Nonetheless, Hückelhoven, Hofer, Coleman and Heß (2018) [46] describe in their review the challenges related to the selection of an FHB-resistant barley being considerably greater compared to wheat due to the high rate of symptomless development of the infection and thus restricting the possibility to select resistant genotypes.

### 2.2. Wheat

In 2019, the European Union countries produced 131.8 million tonnes of wheat, most of it destined for the baking industry (European Commission, 2021). Wheat beer is a special type of beer that is fermented from a mix of malted wheat and barley, where wheat represents at least 50% of the malted cereal, according to the present Provisional Act on Purity of Beer (*Vorläufiges Biergesetz*). The malting procedure of wheat is similar to the one employed for barley; nonetheless, their differences in composition imply significantly different values required for the important malting and brewing parameters (e.g., shorter immersion time during steeping). Wheat contains higher protein and carbohydrate levels (lack of palea husks typical for barley grains); it is rich in arabinoxylans (barley’s most abundant carbohydrate in the endosperm is ß-glucan) and, as a result, compared to barley, it has a higher extract value, greater saccharification power and a lower Kolbach index (KI, soluble nitrogen content as a percentage of total nitrogen content) [47].

As it is in the case of barley, FHB is the main disease affecting wheat fields and has a negative effect on the malting parameters. The coexistence of up to 20 *Fusarium* species might be causing it, each of them having a different mycotoxin production profile: *F. graminearum* and *F. culmorum* produce DON, NIV and ZEN, *F. avenaceum* produces moniliformin (MON) and beauvericin (BEA), *F. poae* mainly produces NIV, T-2 and HT-2 toxins [48]. Besides the production of mycotoxins, the species belonging to *Fusarium* can also produce hydrophobins (that act as surfactants, stabilizing the CO_2_ bubbles in beer) and enzymes that decrease the ß-glucans levels in the brewing wort, affecting its viscosity, increasing the rate of soluble nitrogen that can interfere with the fermentation process and changing wort colour [49]. Habschied et al. (2014) [50] suggested that the infection of *F. culmorum* in wheat can lead to the increase in fungal proteases and activate the plants’ pathogen-related proteins, which are part of its protective strategy (KI of the infected wheat sample 51% versus 49% in control and fungicide-treated samples). Mastanjević et al. (2018) [51] reached a similar conclusion in their study, suggesting that the presence of *F. culmorum* can even influence some of the hereditary traits of the wheat, such as grain hardness (hydrolysis of proteins adhering to the starch granules) and environment-dependent traits, including starch content (starch content decreases with the increase in *Fusarium* infection due to the production of amylolytic enzymes by the fungus), wet gluten content and increased water content, before and after malting compared to a non-infected sample. 

### 2.3. Sorghum

Although barley is the traditional cereal for malting and beer production, its cultivation in tropical areas has not been successful. Thus, to produce beer in these regions barley must either be imported from temperate regions or malting is to be performed with the use of tropical cereals, such as sorghum [52]. Sorghum’s technological quality for malting is primarily defined by its reduced diastatic power (sum of α– and ß–amylase activities), due to the low levels of the ß–amylase contained in sorghum and FAN, which is also lower compared to barley. Malt’s extract (sum of the content of fermentable sugars and unfermentable dextrins) is a less important quality of sorghum malt for conventional opaque beer, as the malt constitutes only 30% of the total cereal grist [53]. Thus, although the technological steps of malting sorghum are the same as for barley (steeping, germination and kilning), the optimal parameters are defined by the abovementioned biological differences of the two kinds of cereals. A review published by Ogbonna (2011) [54] summarizes in detail the challenges and problems related to the malting of sorghum. The author suggests that steeping should last a minimum of 45 h and include an immersion step in warm water (40 °C) to ensure the proper hydration of the kernels. Germination is usually performed at 30 °C for 5 days (barley germination temperature is 16 °C) in an almost saturated atmosphere, and kilning is performed at a maximum 50 °C for 24 h in a forced-draught oven to avoid the denaturalization of the more thermo-sensible sorghum enzymes (maximum barley malt kilning temperature is 80 °C). 

In the case of sorghum cultivars, *Fusarium* infestation in the field is part of a complex disease, sorghum grain mold, which also includes genera such as *Alternaria*, *Phoma* and *Curvularia*. An internal infection of the grain can result in the digestion of starch and protein contained in the endosperm, overall softening and decay of the seed and, most importantly for the quality and safety of the product, the synthesis of the mycotoxins into the caryopsis [55].

Pink, grey, white or black discolourations are the common symptoms visible on the kernels, together with a reduction in grain size, kernel mass and nutritional quality (decrease in soluble carbohydrates and proteins) up to a complete deterioration of the grains [56]. Tesfaendrias, McLaren and Swart (2011) [57] studied the effect of the fungi responsible for the sorghum grain mold on the malting and milling quality of sorghum. Regarding the *Fusarium* genus, they proved that the incidence of *F. proliferatum* and *F. graminearum* can reduce the 1000 kernels weight by 31.47 and 21.26%, respectively; germination yield was reduced by up to 14% in the presence of *Fusarium* spp., and mycotoxins were quantified. The mean concentrations for DON, ZEN and total fumonisins (FBs) analysed by Chilaka, De Boevre, Atanda and De Saeger (2016) [58] in 110 samples of Nigerian sorghum were 100, 38 and 83 µg/kg, respectively. The levels of mycotoxins found during other survey studies were not alarmingly high [59,60], supposedly associated with its high content of tannins and phenols [61]. Nonetheless, the presence of fungi is a threat to the technological quality of the final product to be obtained from sorghum and, being the fifth most-produced cereal in the world, it results in considerable economic losses [62]. 

### 2.4. Other Minor Malting Cereals 

There are several reasons why a brewer would choose to make beer from cereals other than barley, wheat or sorghum. Among them are the local tradition, the increased demand for gluten-free beers for celiac adults, the demand for new organoleptic experiences (e.g., the presence of non-starch carbohydrates and tannins can positively contribute to the mouthfeel) and the production cost (importing barley in tropical regions can be very expensive or even restricted by law). Among these alternative cereals, rye, buckwheat (pseudocereal), oats and millet are malted, all being able to represent up to 100% of the grist for mashing [63]. There are some speciality beers originally produced in Southern Germany (Roggenbier), in which rye malt represents the base of the product (up to 60%) (Wolfe, 2015). Buckwheat (*Fagopyrum esculentum*) is a pseudocereal widely grown in Asia and Eastern Europe. Its use in brewing was only recently discovered due to the absence of glutelin-like proteins, the presence of insoluble starch, antioxidants (rutin) and thus the potential of producing a rich gluten-free beer. The quality parameters of buckwheat malt are poorer compared to barley malt, but they are still within the acceptable range for beer-making [64]. Although obtaining beer from buckwheat malt is not without complications (high wort colour, very slow filtration, need for exogenous enzymes to finish the mashing, higher haziness compared to a wheat beer, etc.), research suggests an optimal relative humidity of the kernels of 35–40% after 7 to 13 h of steeping at 10 °C ensures an acceptable range of malting loss and relatively good quality of malt [65]. Millet malt presents a higher ß–amylase activity and FAN compared to sorghum. Its quality is proved to be directly correlated with the humidity level during germination, which unfortunately also leads to higher malting losses [66]. Oats can also serve as a base for celiac-friendly beers. Although the kernels contain higher amounts of β-glucans, proteins and fats compared to barley (not advantageous for brewing), it is shown that beers obtained from 100% oat malt are characterized by a strong berry flavour, lower alcohol content and higher pH compared to barley beer [67]. Nonetheless, just as in the case of buckwheat, a better yield, additional mouthfeel and granny flavour can be achieved with 10% oat malt as a complement to barley malt beers [68]. 

*Fusarium* species can be typically found in these cereals, and they can produce mycotoxins. Jurjevic et al. (2005) [69] identified the *Fusarium* population in millet grown in Southern Georgia (USA) to be represented by *F. verticillioides*, *F. semitectum*, *F. chlamydosporum* and *F. pseudonygamai*. Later, Leslie and Summerell (2007) [33], in their “The *Fusarium* laboratory manual”, describe *F. thapsinus*, *F. proliferatum*, *F. andiyazi* and *F. pseudonygamai* as the most important species isolated from millet from all the cultivation regions. Rye and oats are also susceptible to FHB, which, similar to the case of the previously mentioned cereals, is mainly driven by *F. graminearum* and less frequently by *F. culmorum*, *F. avenaceum* and *F. poae* [70,71]. During the 2004 Symposium on Buckwheat, Kalinova, Voženilkova and Moudry (2004) [72] reported *F. tricinctum* and *F. avenaceum* being isolated from the surface of buckwheat kernels between 1999 and 2000, among other bacteria. Both species can accumulate mycotoxins in the plants they infect [73]. However, on several occasions, buckwheat seeds extracts were proved to possess antimycotic properties, *Fusarium* spp. being among the target pathogens of these studies [74,75]. 

## 3. *Fusarium* Mycotoxins Transfer from the Cereals to Industrial-Like Beer

From the technological point of view, beer production is considered one of the more complex and delicate processes in the food industry from both a biochemical and physical perspective. It includes steps such as germination, mashing, boiling and fermentation. The question is: can *Fusarium* mycotoxins be transferred from the cereals to the beer? If yes, what is the risk associated with this transfer? As it was described earlier, beer is also prone to contamination with various mycotoxins, which can originate from the malted or unmalted cereals that are used. During the past years, we tried to answer these questions with our research. In the following sections, we will share some of our findings, highlighting the most important stages in malting and brewing, as well as the results obtained by other researchers in the available literature, that have or could have an impact on the levels of *Fusarium* mycotoxins in the case of processing a batch of contaminated barley. The discussion will be focused on the processing steps typical for the beer obtained from pale malt while applying an ale or lager fermentation. 

### 3.1. Malting

Malting is a controlled germination process to produce malt. It consists of three stages (steeping, germination and kilning), which are initiated under specific conditions of humidity and temperature. This is one of the most important production stages for brewing because the quality of the obtained malt will define the quality of the wort and, subsequently, of the beer. aims to create favourable humidity conditions for germination, where the activated enzymes will break starch and proteins. The kilning process inactivates the enzymes before excessive hydrolysis can take place. Additionally, kilning is decisive for flavour and colour formation. 

During our work, we investigated the transfer of DON, DON-3-Glc and ZEN in naturally contaminated and in laboratory-infected barley through the malting process [76]. The effect on the three mycotoxins had a similar tendency and was in accordance with the available studies, proving malting as being a production stage with an impact on fungal infection and mycotoxin contamination. DON was washed out during steeping, registering a reduction of 75 to 85%, which was considerably higher compared to the results obtained by Lancova et al. (2008) [77], who identified a 10% decrease in DON after this step. This reduction could be explained by its solubility in water and by the fact that most of the toxin is located on the outer layers of the kernel, allowing it to be washed out. Vegi, Schwarz and Wolf-Hall (2011) [78] suggested that water flow during steeping could spread the *Fusarium* infection among the grains by 15–90%; however, grain storage has a drastic effect on *Fusarium* viability [79], which makes this danger a very low risk at this processing stage. 

During our studies, after 48 h of germination, an increase in DON concentration up to 30% was observed, followed by a significant decrease in it by the end of it (the resulting concentration was lower compared to the initial one before steeping). An increase in DON-3-Glc levels accompanied this decrease in DON. It could have two origins: first, DON-3-Glc has been released from the matrix thanks to the activation of the hydrolytic enzymes during germination and secondly, it could be formed under the activity of glucosyltransferase [80,81]. 

As was also proved in previous publications [78,82], we identified kilning as not having any destructive effect on either DON or DON-3-Glc. Moreover, we registered an increase in the concentration of both mycotoxins compared to the last day of germination (up to 21.5% increase in DON and up to 107.3% increase in DON-3-Glc). A process with a high potential to reduce the content of DON, according to Kostelanska et al. (2011) [83] is roasting at temperatures above 150 °C. 

There is scarce information concerning the fate of other *Fusarium* toxins during malting, such as ZEN or NIV, which is most likely due to their lower occurrence in barley [84]. We investigated the fate of ZEN during malting and, although some significant fluctuations in its level were observed at different stages of the process, in the end, its level remained unchanged compared to the values identified in the raw barley grains [76].

Geiβinger, Gastl and Becker (2021) [85], in their review, discussed the possibility of the alteration of the metabolism of a barley plant, thus producing a change in the set of enzymes and proteins present in the kernels, due to the pathogen–host interactions that occur during a *Fusarium* infection (e.g., synthesis of an up-regulated ß–amylase, which, besides being in charge of the starch cleavage, also acts as a factor for the programmed cell death of the cells of the grain). Nonetheless, according to the same authors, estimating the impact of the *Fusarium* infestation on the levels of each enzyme is impossible with the current analytical methods because they cannot distinguish between plants’ endogenous enzymes and the ones originating from the fungi unless gene expression studies are conducted. Jin et al. (2021) [86] studied the expansion of hyphal growth and DON fate during malting of FHB infected barley, wheat, rye and triticale grains. The samples of grains that showed a high increase in DON concentration after malting were chosen for the study, even if the original grains contained low levels of the toxin. Their study demonstrates that in barley grains the *Fusarium* hyphae are mainly located on the surface of the husks, but the fungus can colonise the furrow margin of the kernel, which makes it difficult to access during grain cleaning and steeping. In the case of wheat, rye and triticale their imaging techniques showed the presence of the hyphae both on the surface of the kernels and within their interior. These results imply the need for more extensive testing of the malting cereals not only for the presence of mycotoxins but also for the microscopic signs of fungal infection.

### 3.2. Mashing and Boiling

Mashing is the mix of coarse ground malt with a high amount of water under specific temperatures to reactivate all the enzymes present and to allow the conversion of starches into fermentable sugars and of the proteins into amino acids. It aims to ensure a correct fermentation process and achieve the proposed technological quality of the product. Mashing is followed by wort separation and boiling, accompanied by hops addition. The following processes take place during boiling: enzyme and microorganism inactivation, protein precipitation, isomerisation of hop α-acid, evaporation of water and undesirable volatile compounds (e.g., dimethyl sulphides), etc. 

In terms of impact on mycotoxin contamination, in our work [87], mashing and the first 30 min of boiling were proved to have a certain impact. DON and its conjugated forms, ZEN and FBs, were almost entirely transferred from the malt to wort. Moreover, an increase in the extracted amount of toxins was observed through the process, the most significant being registered after 15 min at 45 °C. Wolf-Hall (2007) [88] reported a possible release of DON from the protein complex due to proteolysis during mashing and considering its solubility in water; once released, it would probably pass into the wort. However, regarding ZEN, Inoue, Nagatomi, Uyama and Mochizuki (2013) [89] determined that >80% of it was eliminated with the spent grains. Nonetheless, these variations of the results can be accounted for by the different approaches of introducing the mycotoxins of interest used in the cited study (artificially spiked malt) and ours (laboratory-infected malt with *Fusarium*). Interestingly, a very low incidence of α-zearalenol (α-ZEL) was registered, along with its complete elimination by the end of the process. We also found β-zearalenol (β-ZEL) to be slightly more abundant compared to its stereoisomer, and it showed a low reduction rate at the end of the process. These modified forms of ZEN are produced by the fungi themselves, depending on the prevalent strain [90]. The level of fumonisins during mashing of some samples remained unchanged in the grits (fumonisins concentration in wort < LOQ), while in others it was almost entirely transferred to the wort, as explained by the earlier proved high water solubility of the toxin [14]. 

The first 30 min of boiling had a significant impact on the levels of all mycotoxins, leading to a reduction in their concentration from 90 to 100% compared to the initial level, except for DON where 30 to 60% was detected even after 90 min of boiling. 

A recently published work by Prusova et al., (2022) [91] focused on less studied *Fusarium* mycotoxins, namely nivalenol (NIV), neosolaniol (NEO), enniatins (ENNs), beauvericin (BEA), T-2 and HT-2 toxins during malting and brewing. The obtained results show a significant decrease in all *Fusarium* mycotoxins during steeping (20% NIV, 9% NEO, 18% HT-2, 2% T-2, 33% ENNs and 34% BEA). Nonetheless, germination was characterized by a more than 500% increase in the level of some toxins compared to the levels encountered in the raw materials, which, according to the authors, suggests de novo formation of the toxins by the revived fungi. During brewing, most of the type A trichothecenes was found in the wort, and the subsequent technological steps had no significant impact on their level. Neither ENNs nor BEA was transferred into the wort, being almost entirely found in the spent grains. 

### 3.3. Fermentation

Yeast is critical to the beer-making process and specifically, the fermentation stage. Its activity is not only limited to transforming malt sugars into alcohol, but its enzymes are also crucial in shaping beer flavour and aroma by creating volatile compounds such as esters and fusel alcohols. Two fermentation styles are known worldwide: ale (top fermentation) and lager (bottom fermentation), performed by two different strains of *Saccharomyces* yeast.

Shetty and Jespersen (2006) [92] and Campagnollo et al. (2015) [93] have proved that beer yeasts can bind and metabolise some *Fusarium* mycotoxins during the fermentation process, binding to the cell wall being possible even after the yeast are inactive. In our work, we focused on 15 different *Saccharomyces* strains and 2 *Fusarium* toxins, DON and ZEN [94]. Interestingly, the adsorption dynamics of the two toxins studied are relatively different, most of the adsorbed DON being retained on the yeast cell wall during the first 24 h of fermentation, while ZEN adsorption took place gradually during the 96 h of the process. This difference can be due to various factors such as physical and chemical parameters of the fermentation process (temperature, pH, duration, etc.), the nature of the contamination (natural or spiked) and the different chemical properties of each targeted mycotoxin. The ratios of the observed changes are in line with the available studies [95,96,97], namely from 5 to 15% DON and from 31 to 72% ZEN retained on the yeast cell wall. To identify the role of the viability of the yeast cells in the adsorption process, other studies investigated the ability of brewing yeast residue to adsorb mycotoxins reporting not only the reduction in ZEN (75%) but also that of *Aspergillus* toxins such as aflatoxin B1 (AFB1, 48%) and ochratoxin A (OTA, 59%), due to the β-glucans present in the cell wall [94]. DON was not proved to be efficiently adsorbed by the yeast, reaching a maximum of 17%, which, considering its high occurrence and transfer rate to the wort, may be a subject of concern [98]. However, the study performed by Garda et al. (2005) [99] shows a 53% reduction in DON levels. Studies are available reporting a partial metabolization of the mycotoxins by yeast and the formation of α- and β-ZEL from ZEN [97], and the formation of acetylated-deoxynivalenol and DON-3-Glc from DON [100]. Additionally, we identified a slightly significant difference related to final mycotoxin levels between the fermentation performed by two *Saccharomyces* species (*S. cerevisiae* and *S. pastorianus*), *S. cerevisiae* showing a slightly higher reduction in the levels of the two mycotoxins before and after the process. This can be explained by the production of a higher amount of biomass, which increased the active sites for mycotoxin binding. As was reported previously in the available literature [98], the contamination of the wort with mycotoxins did not have any effect on the biochemical and technological performance of the yeast. Unfortunately, there were no studies found related to the effect of the fermentation on other mycotoxins, such as fumonisins or type A trichothecenes. 

## 4. *Fusarium* Mycotoxins and Craft Beer

The definition of a craft brewery is not uniform around the world; nonetheless, regional and national trade organizations, such as the American Brewers Association, The Brewers of Europe, The German Brewers Association, etc., classify them by production volume, describing them as small, independent and traditional [101]. According to the last report commissioned by The Brewers of Europe, there are 10,300 active breweries owned by more than 9500 brewing companies in the European Union, and with a production of over 405 million hectolitres in 2020, the EU is the second-largest brewing economy in the world after China [102]. Craft beer is the most innovative branch of the brewing industry, and it has seen exponential growth since its emergence in the 1970s in the United States [103]. The innovation can concern the ingredients (use of new blends of grain or rediscovering ancient varieties, opting for organic barley and hops, etc.), the alcohol content (increased demand for low and non-alcoholic beer worldwide creates a demand for new flavours in this product category), brewing steps (use of new technologies, such as high hydrostatic pressure or pulsed light as a less destructive alternative for filtration and pasteurization), barrel ageing, isotonic claims (creating beverages similar to sports drinks with an appropriate osmolality and rich in antioxidants typically present in beer) or packaging (new designs to stand out from the other beers on the shelves) [103]. 

From the perspective of the possible presence of *Fusarium* mycotoxins, the few surveys analysing the presence of different mycotoxins in craft beers suggest that the main contribution to the final level in the product would be made by the malted and unmalted cereals employed in their production. Peters et al. (2017) [104] performed the most extensive survey up to date on the occurrence of different mycotoxins in 1000 beer samples from 47 countries, 60% of which were craft beers. They identified the sum of DON and DON-3-Glc to be above 10 µg/L in 406 samples (40%), 73% of which were craft beers, finding a statistically significant correlation between the %ABV (alcohol by volume) and the toxin concentration. FBs were present in concentrations up to 36 µg/L in the craft beer category of the studied samples, which is still considerably lower compared to the levels reported in traditional African beers, which reached above 1000 µg/L in different areas of the continent [105]. The Imperial Stout beers showed the highest contamination levels in all the analysed mycotoxins with 83% positive samples. Nonetheless, it is a beer style that is hardly consumed, even by craft beer enthusiasts, mostly due to its higher alcohol content, price and lower accessibility (complex technological steps require a more unique state of the art and knowledge). The hop-forward beer styles are the most popular among the consumers, representing above 20% of the world craft beer production, among them Indian Pale Ale (IPA), Imperial IPA and New England IPA (NEIPA) with bitterness ranging from 30 to 100 IBU (international bitterness units) [103] Although there are many original research articles dedicated to the improvement of the production technology of the craft beers (605 Scopus document search results, 30th of September 2021), no published works have studied the fate of *Fusarium* mycotoxins during brewing or surveying their levels in malts and hops used in production. 

Many customers choose products originating from organic agriculture [106]. Beer is also on this list, and more craft breweries around the world are working to provide this option. There are no studies on the levels of mycotoxins in organic beers; nonetheless, the surveys comparing organic and conventional barley, wheat and oats showed that the organic cereals contained less *Fusarium* infestation and lower trichothecenes levels compared to the ones obtained by conventional agriculture [107]. Pleadin et al. (2017) [108] also assessed the occurrence of *Fusarium* mycotoxins in organic and conventional cereals from Croatia but did not find significant differences in the content of the mycotoxins between the two types. The fact that organic agriculture shows similar or lower levels of fungal infestation could be related to the field practices applied, such as crop rotation, soil management, field density, biocontrol agents or others, that were proved to be a reliable strategy in fungal and mycotoxin mitigation in the field [36].

As mentioned above, craft beer is also a window of opportunity for the application of emerging processing technologies to ensure the safety of the product without interfering with its organoleptic profile. Mirza Alizadeh et al. (2021) [109] reviewed the impact that technologies such as cold plasma, pulsed light, ultrasound, pulsed electric field or high-pressure processing could have as eco-friendly and economical methods in fungal and mycotoxin mitigation. Most of them were proved efficient in reducing the fungal count in solid foods and cereals (up to 93% or even 100% in the case of cold plasma), but the results on mycotoxins are rather limited, and no studies have focused on beer or its intermediate products. 

## 5. Concluding Remarks and Future Perspectives

*Fusarium* species are an important threat to cereal production. For the cereals typically used in beer production, *F. graminearum*, *F. poae*, *F. proliferatum* and *F. verticillioides* are the most relevant in terms of frequency of occurrence and mycotoxin production. They can cause yield and quality losses in the crops, to the extent of making them completely unusable in the malting or brewing processes. During malting, several studies suggest an increase in trichothecenes production occurs, either due to the kernels’ enzymatic activity or de novo production by the fungi surviving in the interior of the grains. The hydrophilic nature of the *Fusarium* mycotoxins allows them to be transferred from the malted or unmalted cereals to the sweet wort and then beer. The fermentation by *Saccharomyces* yeasts has been repeatedly proved as being able to adsorb < 17% DON and <70% ZEN or metabolise part of the mycotoxins into their respective modified forms. Craft beers have been less studied in terms of mycotoxin occurrence and consumption-associated risk. Their market is growing and so is the innovation that each small brewer brings to the table. The surveys investigating the occurrence of *Fusarium* mycotoxins in the craft beer identified a direct correlation between the %ABV and the concentration of DON and DON-3-Glc [104,110], probably due to the use of a higher amount of malted and unmalted cereals to provide additional substrate for the fermentation. Due to the high proportion of water to the quantity of malt grist and unmalted cereals, the levels of mycotoxins in the product are considerably lower and do not represent a risk for an occasional consumer. Nonetheless, considering that there are no maximum limits of mycotoxins for beer established by the EU regulation, the control of the raw materials is of crucial importance to ensure not only a safe but also enjoyable drink. 

In terms of future perspectives, several aspects require more research. Recent studies investigate the possibility of alternative yeast genera that can be domesticated to be used in brewing to provide a more innovative approach to product formulation by generating new flavours and styles [111], such as *Brettanomyces*, *Torulaspora*, *Lachancea*, *Pichia*, *Mrakia*, etc. Nevertheless, their potential to reduce often occurring mycotoxins was not yet researched. Some of the new technologies such as cold plasma and pulsed light are proved to have the in vitro potential to reduce fungal count and modify mycotoxins; however, more realistic experiments need to be planned to obtain a better understanding of the nature of these changes.

## Figures and Tables

**Figure 1 foods-11-00246-f001:**
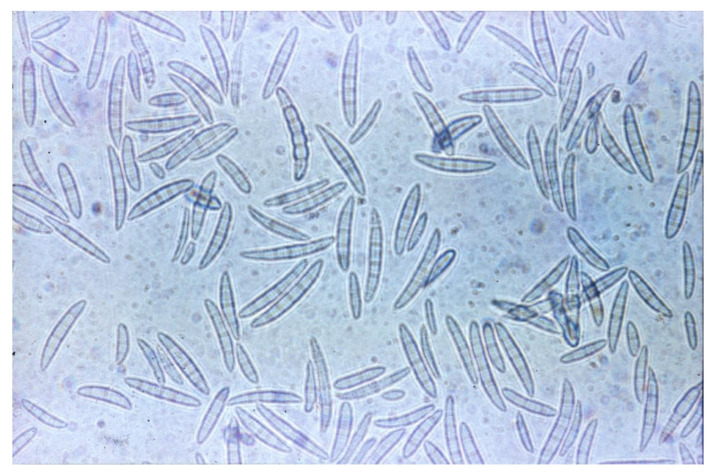
Macroconidia of *Fusarium* sp.

**Table 1 foods-11-00246-t001:** Toxicity of *Fusarium* mycotoxins relevant for malt and beer.

Mycotoxin Group	Relevant Representatives	Producing Fungi	Most Affected Cereals	Toxicity in Humans and Animals	References
Trichothecenes A	T-2 and HT-2 toxins	*F. sporochioides*, *F. langsethiae*	Oats, barley	Hepatotoxicity, decrease in cell viability, inhibition of cell proliferation, oxidative stress, mitochondria damage, alimentary toxic aleukia (ATA), disruption of DNA and RNA synthesis	[20,21]
	DAS	*F. equiseti*	Wheat, oat barley, rye, sorghum	Immunotoxicity, hematotoxicity, pulmonary and growth disorders, gastrointestinal lesions and diarrhea observed in various farm animals	
Trichothecenes B	Nivalenol	*F. graminearum*	Wheat, rye	Immunotoxic, genotoxic, disruption of microbial homeostasis, development of chronic enteric disease	[22,23,24]
	DON, DON-3-Glc, 3- and 15-AcDON	*F. graminearum*, *F. culmorum*, *F. cerealis*	Wheat, barley, maize, oat, rye	Alterations of intestinal structures, disruption of epithelial barriers, impairment of intestinal mucosal immune response, changes in gut microbiota composition, growth retardation	
Zearalenone	ZEN, α-ZEL, ß-ZEL, etc.	*F. graminearum*	Maize	Estrogenic effect, DNA methylation, decrease in embryo implantation rate, oxidative stress, decreased testosterone concentration and increased progesterone level	[25]
Fumonisins	FB1, FB2, FB3, FB4	*F. verticillioides*, *F. proliferatum*	Maize	Disruption of sphingolipid metabolism, oesophageal and liver cancers, neural tube defects, cardiovascular problems	[26]
Emerging mycotoxins	Beauvericin and enniatins	*G. fujikuroi* complex	Wheat, oat	Cytotoxic, potential genotoxic, hematotoxic	[27]
	Butenolide	*F. graminearum* *F. equiseti*	Wheat, oat, barley, rye, sorghum	Inhalation toxicity, dermal toxicity, cytotoxicity, potential induction of myocardial damage	[28,29,30]
	Fusarin C	*F. verticillioides*, *F. graminearum*	Maize	Cancerogenic (oesophageal and breast), mutagenic, cytotoxic	[31]
	Equisetin	*F. equiseti*, *F. semitectum*	Wheat, oat, barley, rye, sorghum	Moderate toxicity to mice	[22]
	Neosolaniol	*F. graminearum*	Barley, maize, rice, sorghum, wheat, triticale	Anorectic response to exposure in mice (stronger in the case of an intraperitoneal than oral exposure)	[21,32]

DAS = diacetoxiscirpenol; DON = deoxynivalenol; DON-3-Glc = deoxynivalenol-3-glucoside; 3-AcDON = 3-acetyldeoxynivalenol; 15-AcDON = 15-acetyldeoxynivalenol; α-ZEL = α-zearalenol; ß-ZEL = ß-zearalenol; FB = fumonisin (B_1_, B_2_, B_3_ and B_4_).

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
