# Peer review of "Relevant *Fusarium* Mycotoxins in Malt and Beer"

_foods, 2022, doi:10.3390/foods11020246_

Round 1
Reviewer 1 Report
The manuscript reviews the issue of Fusarium mycotoxins in beer. This is a significant subject, given the popularity of beer as an alcoholic (and increasingly often - non-alcoholic) beverage, a potential infestation of barley, wheat, sorghum and other cereals used in beer production and eventually the toxicity of Fusarium metabolites. The manuscript entirely fits into the aims and scope of Foods. It is written in good English language. However, there are some major points to be addressed first to strengthen the message of this work, improve its outreach and impact
- Do you have any microscopic images of Fusarium species characterized in the manuscript that would accompany the text nicely?
- Provide a table summarizing the observed levels of Fusarium-related mycotoxins in different beers - this would be a great addition to the manuscript and its impact. It would be great to distinguish between craft and non-craft beers.
- Provide key recommendations on mitigating the presence of Fusarium mycotoxins in beer on different stages of production in a "bench-to-bed" manner. It will increase the impact of the manuscript.
- It would be good to provide a summary of the toxicity of Fusarium mycotoxins. Provide a separate section and a table showing the mechanism of action/outcome of exposure to particular Fusarium mycotoxins, the presence of which is plausible in beer.
- It would be best to provide a risk assessment given the average beer consumption, e.g., in Europe. How do the noted concentrations translate into the health risks? This aspect needs to be discussed more closely.
Other comments:
- L9: many times -> has been numerously reported.
- L11: present -> characterize...(...) that could infect
- L17: explain all abbreviations at first mention; since this is the only mention of these mycotoxins within an Abstract, just write their full name.
- L17: This sentence is not entirely clear: slight reduction in DON, but high of ZEN? Rephrase for clarity.
- L26-27: Name of first described should be given, e.g., Fusarium graminearum Schwabe
- L28: Reformulate to indicate how old it is because this statement is confusing (e.g., there are many traditions).
- L30: Hordeum vulgare L. (at first mention, provide a describer - you don't need to do it later, vide L78).
- L56: to their detection -> for their detection.
Author Response
Dear Reviewer,
Thank you for the comments and suggestions that will surely improve the quality of the manuscript. All of them were considered and the respective changes were made. Please find bellow the point-by-point responses.
The manuscript reviews the issue of Fusarium mycotoxins in beer. This is a significant subject, given the popularity of beer as an alcoholic (and increasingly often - non-alcoholic) beverage, a potential infestation of barley, wheat, sorghum and other cereals used in beer production and eventually the toxicity of Fusarium metabolites. The manuscript entirely fits into the aims and scope of Foods. It is written in good English language. However, there are some major points to be addressed first to strengthen the message of this work, improve its outreach and impact
- Do you have any microscopic images of Fusarium species characterized in the manuscript that would accompany the text nicely?
A microscopic image of Fusarium species was added to the manuscript.
- Provide a table summarizing the observed levels of Fusarium-related mycotoxins in different beers - this would be a great addition to the manuscript and its impact. It would be great to distinguish between craft and non-craft beers.
We agree with you that indeed a table could be a good addition to the paper, nonetheless it was published several years ago and in this work we included the two new studies that were performed since then, while also presenting the reference to that table.
- Provide key recommendations on mitigating the presence of Fusarium mycotoxins in beer on different stages of production in a "bench-to-bed" manner. It will increase the impact of the manuscript.
Key points on the mitigation of Fusarium mycotoxins during beer production were added to the last section of the article.
- It would be good to provide a summary of the toxicity of Fusarium mycotoxins. Provide a separate section and a table showing the mechanism of action/outcome of exposure to particular Fusarium mycotoxins, the presence of which is plausible in beer.
It is not entirely the subject of the present review. Also, we find that there are many published works that reviewed the toxicity of the mycotoxins typically found in beers. We included a particular mention with the respective reference.
- It would be best to provide a risk assessment given the average beer consumption, e.g., in Europe. How do the noted concentrations translate into the health risks? This aspect needs to be discussed more closely.
We agree with you that a risk assessment of the exposure to the regulated mycotoxins through beer consumption would be a great contribution to the outreach of the paper. Nonetheless, the data available on beer consumption is usually annual, thus it is difficult to isolate a daily consumption pattern and thus, obtain a reliable evaluation.
Other comments:
- L9: many times -> has been numerously reported.
- L11: present -> characterize...(...) that could infect.
- L17: explain all abbreviations at first mention; since this is the only mention of these mycotoxins within an Abstract, just write their full name.
- L17: This sentence is not entirely clear: slight reduction in DON, but high of ZEN? Rephrase for clarity.
- L26-27: Name of first described should be given, e.g., Fusarium graminearum Schwabe.
- L28: Reformulate to indicate how old it is because this statement is confusing (e.g., there are many traditions).
- L30: Hordeum vulgare L. (at first mention, provide a describer - you don't need to do it later, vide L78).
- L56: to their detection -> for their detection.
All the specific comments were addressed, and the changes performed in the manuscript.
Reviewer 2 Report
The manuscript entitled “What everyone must know about Fusarium mycotoxins in beer” is a review aimed at describing and discussing the up-to-date available knowledge about Fusarium mycotoxins along the beer supply chain. After an overall introduction of the topic, the contamination by Fusarium species has been reported for all the brewing cereals, including minor malting cereals as millet, rye, oats and buckwheat. Furthermore, Fusarium mycotoxin transfer from cereals to beer during the different steps of the brewing process has been detailed and discussed. A specific paragraph has been spent for craft beer, being this product of growing interest for consumers. Finally, the paragraph with concluding remarks reports the main findings described in the paper, summarizing also the most promising topics worth studying in the next future regarding brewing.
This manuscript addresses a very interesting topic, providing, comparing and discussing several studies and investigations really useful for brewers and researchers.
The paper is very detailed and well organized, being appropriate also the choice of the paragraphs. Unfortunately, it needs to be heavily improved in writing. The manuscript should be rewritten by using the same structure, but writing sentences more accurately making them more readable.
Since a deep rewriting of the text is required, the punctual and numerous corrections of the whole text have not been reported in this revision report. Therefore, only a few general comments are given below.
In line 311, the authors should explain which are the different approaches they refer to.
One suggestion is to replace the title of the paragraph 2.4 (“Less frequently used cereals”) with “Other minor malting cereals”.
Further suggestions are to revise some references, for example in line 65 replace the bibliographic citation “Arie, 2019” with “Leslie and Summerell, 2006”. About this reference, the year reported in lines 64 and 612 (2007) should be corrected in 2006.
More, all bibliographic citation in the text need to be reported as numbers in brackets, formatted according the journal’s instruction for authors.
Author Response
Dear Reviewer,
Thank you for taking the time and reviewing the paper. We considered all the comments and did our best to make all the necessary editing of the manuscript. Please find below the point-by-point responses.
manuscript entitled “What everyone must know about Fusarium mycotoxins in beer” is a review aimed at describing and discussing the up-to-date available knowledge about Fusariummycotoxins along the beer supply chain. After an overall introduction of the topic, the contamination by Fusarium species has been reported for all the brewing cereals, including minor malting cereals as millet, rye, oats and buckwheat. Furthermore, Fusarium mycotoxin transfer from cereals to beer during the different steps of the brewing process has been detailed and discussed. A specific paragraph has been spent for craft beer, being this product of growing interest for consumers. Finally, the paragraph with concluding remarks reports the main findings described in the paper, summarizing also the most promising topics worth studying in the next future regarding brewing.
This manuscript addresses a very interesting topic, providing, comparing and discussing several studies and investigations really useful for brewers and researchers.
The paper is very detailed and well organized, being appropriate also the choice of the paragraphs. Unfortunately, it needs to be heavily improved in writing. The manuscript should be rewritten by using the same structure, but writing sentences more accurately making them more readable.
Thank you for the comments and suggestions. The paper was proofread again and the necessary grammar corrections were made.
Since a deep rewriting of the text is required, the punctual and numerous corrections of the whole text have not been reported in this revision report. Therefore, only a few general comments are given below.
In line 311, the authors should explain which are the different approaches they refer to.
Corrected
One suggestion is to replace the title of the paragraph 2.4 (“Less frequently used cereals”) with “Other minor malting cereals”.
Replaced
Further suggestions are to revise some references, for example in line 65 replace the bibliographic citation “Arie, 2019” with “Leslie and Summerell, 2006”. About this reference, the year reported in lines 64 and 612 (2007) should be corrected in 2006.
Corrected
More, all bibliographic citation in the text need to be reported as numbers in brackets, formatted according the journal’s instruction for authors.
Corrected
Round 2
Reviewer 1 Report
Author's only partially responded to my comments. Importantly, they refused to discuss the toxicity relevant to the subject (shouldn't the toxicity of Fusarium toxins be an integral part of what "everyone should know"?). They also refused to addrrss the risk assessment for beer consumption which is possible using approximate estimates.
Author Response
Dear Reviewer,
Thank you very much for your comments. We have followed your remarks and added a table summarizing the toxicity of Fusarium mycotoxins. A paragraph in the introduction section was added regarding the contribution of the consumption of beer to the TDI and PMTDI for relevant Fusarium mycotoxins, citing a detailed recently published systematic review and meta-analysis.
Kind regards,
Reviewer 2 Report
In my first step of revision I explained that the manuscript was well organized, but it needed to be heavily improved in writing. I suggested to rewrite the manuscript by using the same structure, but writing sentences more accurately making them more readable. However, the authors replied only to minor comments, while the manuscript was left almost untouched, except for about two sentences.
Author Response
Dear Reviewer,
Thank you for your remarks and comments. We followed your suggestions and performed an in-depth grammar and language check of the manuscript.
Kind regards.